# Impact of Soil Chemical Properties on the Growth Promotion Ability of *Trichoderma ghanense**, T*. *tomentosum* and Their Complex on Rye in Different Land-Use Systems

**DOI:** 10.3390/jof8010085

**Published:** 2022-01-15

**Authors:** Danguolė Bridžiuvienė, Vita Raudonienė, Jurgita Švedienė, Algimantas Paškevičius, Ieva Baužienė, Gintautas Vaitonis, Alvyra Šlepetienė, Jonas Šlepetys, Audrius Kačergius

**Affiliations:** 1Nature Research Centre, 08412 Vilnius, Lithuania; danguole.bridziuviene@gamtc.lt (D.B.); vita.raudoniene@gamtc.lt (V.R.); algimantas.paskevicius@gamtc.lt (A.P.); ieva.bauziene@gamtc.lt (I.B.); gintautas.vaitonis@gamtc.lt (G.V.); 2Lithuanian Research Centre for Agriculture and Forestry, 58344 Akademija, Lithuania; alvyra.slepetiene@lammc.lt (A.Š.); jonas.slepetys@lammc.lt (J.Š.); audrius.kacergius@lammc.lt (A.K.)

**Keywords:** growth promotion, *Trichoderma*, physiological characteristics, CM-activity of soil

## Abstract

Microbial-based biostimulants that increase plant performance and ensure sustainable restoration of degraded soils are of great importance. The aim of the present study was to evaluate the growth promotion ability of indigenous *Trichoderma ghanense*, *T. tomentosum* and their complex on early rye seedlings in sustained grassland and arable soil. The impact of soil chemical properties on the ability of selected *Trichoderma* strains and their complex to promote plant growth was determined by the evaluation of the rye (*Secale cereale* L.) early seedling growth—measuring the length of shoots and roots as well as their dry weight. *Trichoderma* species were tested for their ability to produce extracellular degradative enzymes on solid media. Furthermore, the soil properties and CM-cellulase activity of soil were estimated. The indigenous *Trichoderma* strains possess the capacity to produce enzymes such as peroxidase, laccase, tyrosinase, and endoglucanase. The results indicated a significant (*p* < 0.05) increase in plant growth and the improvement of some soil chemical properties (total N, mobile humic and fulvic acids, exchangeable K_2_O, soil CM-cellulase activity) in inoculated soils when compared to the control. The growth of the roots of rye seedlings in sustained grassland was enhanced when *T. tomentosum* was applied (*p* = 0.005). There was an increase in total weight and shoot weight of rye seedlings when *T. ghanense* was used in the arable soil (*p* = 0.014 and *p* = 0.024). The expected beneficial effect of *Trichoderma* spp. complex on rye growth promotion was not observed in any tested soil. The results could find application in the development of new and efficient biostimulants, since not only do physiological characteristics of fungi play an important role but also the quality of the soil has an impact.

## 1. Introduction

Intensive farming, the abundant use of fertilizers—especially nitrogenous—and pesticides, and the prevalence of monocultures and intensive tilling have an unavoidable negative influence on soil microbiota and crop yield worldwide [1]. Microorganisms are responsible for the decomposition of organic matter in the soil and involved in the formation of humus, the improvement of soil quality and the supply of nutrients to plants. The demand for food for the growing human population is constantly increasing; the world will need 70–100% more food by 2050. The production of cereals, especially wheat, rice, and maize, which account for half of human calorie intake, has to be increased [2]. Indeed, more sustainable agriculture requires yield increases and improved product quality, while reducing the negative impact of agrochemicals on the environment, all of which might be fostered by microbial-based biostimulants [3]. Microbial-based biostimulants have been important since the beginning of agriculture (i.e., *Rhizobium* in legumes). These biostimulants or biofertilizers increase plant performance and ensure the sustainable restoration of degraded soils [3,4,5,6]. Recently, biofertilizers have been produced from microorganisms that are able to mineralize organic matter in the soil or accumulate inorganic matter from the atmosphere, thus providing plants with nutrients. Moreover, some of them show plant growth promotion and biocontrol abilities [5,7,8].

Fungi such as *Trichoderma* are often used as growth promotion agents and biofertilizers. Extensive research has been focused on their physiological and biological characteristics. They produce enzymes such as cellulase complex enzymes, xylanase, chitinase, amylase, and pectinase that enable the efficient decomposition of organic matter in the soil, thus providing plants with nutrients [9,10,11,12]. According to López-Bucio et al. [3], the best studied *Trichoderma* species regarding their mechanisms of action are *T. asperullum*, *T. artoviride*, *T. harzianum*, *T. virens* and *T. viride*, most of which also exhibit high biostimulant action on horticultural crops. The extracellular oxidoreductases produced by *Trichoderma* participate in the decomposition of phenolic compounds of both natural and xenobiotic substances, resulting in the object of soil bioremediation agents [13]. High growth rates and strong antagonistic abilities that are characteristic of many *Trichoderma* spp., allow their use as biocontrol agents against pathogenic fungi such as *Botrytis*, *Fusarium*, *Rhizoctonia*, *Sclerotinia,* etc. [7,14,15]. Moreover, fungi such as *Trichoderma* promote the growth of plant roots and shoots by solubilization of phosphates and micronutrients in soil [8,16]. Furthermore, there are reports on their ability to enhance plant tolerance to environmental stresses, such as drought or high salinity [17,18,19].

All of these characteristics make *Trichoderma* spp. a very promising agent to produce *Trichoderma*-based biostimulant products for the sustainable management of agriculture [20]. Much work has been invested in different technologies for biofertilizers, biopesticides, and biostimulants, as well as studies of their mechanisms [5,21,22]. Several *Trichoderma*-based product application methods have been developed for seeds, seedlings, plants, or soil treatment. Most *Trichoderma*-based products are used as biopesticides and little attention has been paid to *Trichoderma* as a biofertilizer or plant growth promoter. [23]. Moreover, not much study has been focused on inefficient products after field treatment [24]. The production of metabolites and the activity of enzymes of *Trichoderma* greatly depends on the environmental conditions. Therefore, the impact of *Trichoderma*-based products is expected to be different in soils of varying quality. A complex of soil environment properties unsuitable for the development of *Trichoderma* could explain cases where the use of fungal inoculants was inefficient [21]. Caporale et al. [25] discovered the different effects of arable, grove, and forest soils on the efficiency of two *T. harzianum* strains in promoting the growth of *Brassica rapa*. However, there have not been enough studies on the effect of *Trichoderma* spp. on plant growth promotion in different land-use systems. Soil is a variable and labile system; therefore, the fungi with appropriate physiological and biological characteristics are important for the creation of efficient bio-products. For that reason, deep studies of fungi in soils of varying quality are required. The aim of the present study was to evaluate the growth promotion ability of indigenous *T. ghanense*, *T. tomentosum* and their complex species on early rye seedlings in sustained grassland and arable soil.

## 2. Materials and Methods

### 2.1. Fungi

Two indigenous *Trichoderma* strains were used in this research. These two strains were isolated from winter rye (*Secale cereale*) rhizospheric soil of the Vokė Branch of the Lithuanian Research Centre for Agriculture and Forestry (Vilnius, Lithuania) during the summer season of 2015. For *Trichoderma* isolation, the soil serial dilution plate method was used [26]. In brief, 10 g of soil sample was added into 90 mL of sterilized water and shaken on an orbital shaker at 200 rpm for 1 h. Immediately after shaking, a series of 10-fold dilutions of the suspension were carried out, and appropriate dilutions were plated on malt extract agar with chloramphenicol (250 mg L^−1^). The plates were incubated at 28 °C for 5 days. Individual colonies of fungi were isolated, purified and maintained on potato dextrose agar (PDA; Oxoid, Basingstoke, Hampshire, UK) slants at 4 °C.

### 2.2. Identification of Fungi

The identification of fungi was based on morphological characteristics, internal transcribed spacer (ITS) region and translation elongation factor 1 alpha (TEF1) gene sequence analysis [27,28].

The microscopy observation of *Trichoderma* spp. was performed using a Leica DM5000 microscope with a Leica DFC450 camera mounted on top. *Trichoderma* spp. morphology was examined from cultures grown on MEA at 28 °C for 5 days.

For genomic DNA, approximately 100 mg wet weight of mycelium was extracted from fresh cultures grown on malt extract agar (MEA, Liofilchem, Roseto d. Abruzzi (TE), Italy) with ZR Fungal/Bacterial DNA MiniPrep™ Kit (Zymo Research Europe GmbH, Freiburg im Breisgau, Germany) following the manufacturer’s instructions. The internal transcribed spacers 1 and 2 of rDNA, including the 5.8S rDNA, were amplified using primers ITS5 [29] and AscoITS4 [30], and the region of translation elongation factor 1α (TEF1) was amplified using primers EF-1 and EF-2 [31].

The PCR was performed in a Veriti-96 Gradient Thermocycler (Applied Biosystems, Waltham, MA, USA) in 20 µL volume applying 10 µL KAPATaq Ready Mix (KAPAbiosystems, Wilmington, DE, USA) according to the manufacturer’s instructions. Sequencing of purified PCR products was performed by BaceClear B.V. (Leiden, The Netherlands). From each specimen, two different PCR products were sequenced in both directions (5′ and 3′) using the same primers used in the amplification reactions. The Basic Local Alignment Sequence Tool for nucleotides (BLASTn) search program (NCBI) performed the rDNA homology searches. All sequences are available at the National Center for Biotechnology Information (NCBI) GenBank under the accession numbers OL892052/MW829414 and OL892051/MW829415. The phylogenetic analyses were conducted in MEGA 11 [32] using the maximum parsimony method applying the subtree-pruning-regrafting (SPR) algorithm with search level 1, in which the initial trees were obtained by the random addition of sequences (5 replicates). The trees were drawn to scale, with branch lengths calculated using the average pathway method and were in the units of the number of changes over the whole sequence [33]. The percentage of replicate trees in which the associated taxa clustered together in the bootstrap test (1000 replicates) are shown next to the branches [34].

### 2.3. Trichoderma spp. Physiological Characteristics

*Trichoderma* fungi were tested for their ability to produce extracellular degradative enzymes (peroxidase, laccase, tyrosinase, and endoglucanase) on solid media. Additionally, the change in lignin and cellulose content in the straw was recorded.

Flasks of 50 mL were filled with 0.5 g of the rye (*Secale cereale*) straw, which was grounded into particle sizes of 2–4 mm and sterilized by autoclave. After sterilization, the straw was moistened with a sterile mineral solution (3 mL per flask). Mineral solution: NH_4_NO_3_ 3 g L^−1^ and KH_2_PO_4_ 1 g L^−1^. Each flask was inoculated with a 9-mm disc of fungal mature culture (one disc per flask), and solid-state fermentation was maintained at 28 °C for 30 days. The flasks without fungi were used as a control. All experiments were performed in triplicate.

Enzymatic assay of peroxidase (E.C. 1.11.1.7) was based on the colorimetric evaluation of the oxidation product of o-dianisidine in the presence of H_2_O_2_ [35]. The reaction mixture contained 0.5 mL of cultural liquid, 3 mL of o-dianisidine reagent (50 mL 0.4 M phosphate buffer (pH 5.9), 2 mL of 1% o-dianisidine, 200 mL of distilled water), and 0.2 mL of 0.05% H_2_O_2_. Tests and controls were incubated at 20 °C for 5 min in a water bath. The reaction was stopped by addition of 50% H_2_SO_4_. Absorbance of the reaction mixtures was measured at 560 nm using an Evolution 60S spectrophotometer (ThermoFisher Scientific, Waltham, MA, USA). The activity of peroxidase was calculated according to the coefficient of micromolar extinction, with the value of 0.0128. Peroxidase activity was expressed as activity units (a.u.) g^−1^.

Enzymatic assay of laccase (E.C. 1.10.3.2.) was conducted according to Ravin and Harward [36] methodical recommendations. The reaction mixture contained 0.1 mL of cultural liquid, 1 mL 0.5% of p-phenylenediamine hydrochloride, and 2 mL 0.1 N of acetate buffer (pH 6). The reaction was stopped by adding 1 mL 0.1% of sodium azide solution. Absorbance of the reaction mixtures was measured at 530 nm using an Evolution 60S spectrophotometer (ThermoFisher Scientific, Waltham, MA, USA). Laccase activity was expressed as activity units (a.u.) g^−1^.

Enzymatic assay of tyrosinase (E.C. 1.10.3.1.) measured absorbance spectrophotometrically using a method based on the estimation of the optical density of reaction products formed during oxidation of pyrocatechin over a given period [37]. The reaction mixture contained 0.5 mL of cultural liquid, 2 mL 0.06 M of phosphate buffer (pH 7.4), and 0.5 mL 0.05 M of pyrocatechin. Absorbance of the reaction mixtures was measured at 420 nm using an Evolution 60S spectrophotometer (ThermoFisher Scientific, Waltham, MA, USA). Indications of the spectrophotometer were recorded every 20 s for 2 min. Tyrosinase activity was expressed as conditional units (c.u.) g^−1^.

Enzymatic assay of endoglucanase (E.C. 3.2.1.4) was estimated with Na-carboxymethylcellulose (Na-CMC) [35]. One ml of 1% Na-CMC was placed in a test tube containing 1 mL of a cultural liquid and incubated at 40 °C for 30 min. After incubation, endoglucanase activity was estimated according to the amount of the reducible substance in 1 mL of the reaction compound using an o-toluidine reagent. The reaction mixture containing 1 mL of cultural liquid and 7 mL of o-toluidine reagent was boiled for 10 min. The amount of glucose was measured at 560 nm with an Evolution 60S (ThermoFisher Scientific, Waltham, MA, USA). The amount of reducible substance was estimated according to the glucose calibration curve. Endoglucanase activity was expressed in percent (%).

The content of lignin in the rye straw was evaluated following the method of Chudiakova [38]. For the estimation of lignin degradation, after incubation, 0.5 g of rye straw was placed in 300 mL flasks containing 60 mL of 2% HCl and was boiled for 2 h. The content was filtered, washed with water until the acid reaction ceased, and then was washed with acetone until the filtrate became clear. The rye straw was dried and transferred into 7 mL of 72% H_2_SO_4_. Hydrolysis was continued for 2.5 h at 20 °C with periodic shaking. Then, 93 mL of water was added into each flask and the content boiled for 1 h after fitting with a return condenser. After filtration, the filtrate was washed with NaCl solution (0.5 g L^–1^) to eliminate acid. The filter with lignin was dried at 105 °C for 4 h and weighed. The loss of lignin (%) was determined by comparison to the corresponding content in the control rye straw.

The content of cellulose in the rye straw was evaluated following Kürschner’s and Hafer’s method [37]. The method is based on the oxidation, decomposition, and dissolution of various phenolic compounds of the plant. Cellulose remains unchanged. After incubation, 0.5 g of rye straw was placed in 300 mL flasks containing a 30 mL mixture of HNO_3_ and C_2_H_5_OH (using ratio 1:4) and boiled for 1 h. The content was filtered and washed with pure ethanol. After filtration, the rye straw was dried and transferred into flasks containing 50 mL of 0.3 M NaOH and boiled for 1 h after fitting with a return condenser. After filtration, the filtrate was washed with H_2_O, dried at 105 °C for 4 h, and weighed. The loss of cellulose (%) was determined by comparing the corresponding content in the control rye straw.

The colony growth rates of *Trichoderma* spp. at different pH (4; 5; 6; 7 and 8) values were tested on Czapek Dox agar (CzDA, Liofilchem, Italy) and at several temperatures—5, 15, 26, and 35 °C—on PDA. The mycelial growth was measured after 5 days of incubation. All experiments were performed in triplicate.

### 2.4. Soil Sampling and Analysis

Samples taken from the top layer (0–25 cm) of soil developed by two different land-use systems—soil from sustained grassland (1S) and arable soil (crop rotation field) (2S)—were used for the experiment. Both investigated topsoil horizons were identified as Eutric Endocalcaric Endostagnic Cambisol on light loam of glacial moraine.

Grassland 1S was a pasture with a mixture of grasses and leguminous herbs sown in 1945. The pasture has been used for 74 years without reseeding. Currently, the pasture-formed grassland is semi-natural, dominated by grasses.

Arable soil 2S had been in ecological crop rotation since 2003. Crop rotation three-field: 1—winter wheat, 2—summer barley with red clover under sowing, and 3—red clover for seed. The soil sample was collected after winter wheat cultivation.

Before the experiment, the chemical properties and CM-cellulase activity of the original soil samples were evaluated. The chemical analysis of soil was performed at the Chemical Research Laboratory of the Lithuanian Research Centre for Agriculture and Forestry (Table 1).

During the experiment (the original soil and soil after inoculation), the number of fungi was determined by the soil serial plate method [26]. The diluted samples were directly plated onto malt extract agar with chloramphenicol (250 mg L^−1^). Plates were incubated at 26 °C for 2–5 days, and the total colony-forming units (CFU) of each repetition were counted. All experiments were repeated in triplicate.

### 2.5. The Estimation of CM-Cellulase Activity of Soil

Ten grams of field-moist soil were placed into three 100 mL flasks. Then, 15 mL of 0.7% Na-CMC and 15 mL of 2 M of acetate buffer were added to two flasks (samples). For the control, only 15 mL of acetate buffer was added into the third flask. The flasks were incubated at 50 °C for 24 h. The samples were filtered immediately after incubation, and 0.5 mL of filtrates were diluted to 20 mL with distilled water in test tubes. One ml of diluted filtrate, 1 mL of reagent A (16 g of anhydrous sodium carbonate and 0.9 g of potassium cyanide in 1000 mL of distilled water), and 1 mL of reagent B (0.5 g of potassium hexacyanoferrate (III) in a 1000 mL of distilled water) were added into a test tube, sealed, and incubated for 15 min in a boiling water bath.

After cooling, 5 mL of reagent C (1.5 g of ferric ammonium sulphate, 1 g of sodium dodecyl sulphate, and 4.2 mL of concentrated H_2_SO_4_ in 1000 mL of distilled water) was added. Absorbance of the reaction mixtures was measured at 690 nm using an Evolution 60S spectrophotometer (ThermoFisher Scientific, Waltham, MA, USA). CM-cellulase activity of soil was expressed as µg of glucose equivalents (GE) per gram dry matter and incubation time (µg GE g^−1^ · dm · 24 h^−1^) [42].

### 2.6. Design of Experiment

Plant pots (11 cm in diameter, 0.5 L volume) were filled loosely with non-sterilized grassland 1S or arable soil 2S. For inoculum, the fungi were cultivated on PDA at 26 °C for 7 days. Suspensions of *Trichoderma* spp. were prepared in 0.9% saline from fully mature cultures. The concentration of the suspensions was determined by measuring the optical density at 530 nm with an Evolution 60S (Thermo Fisher Scientific, Waltham, MA, USA) and then checked by plating on PDA. The final inoculum concentrations were 1 × 10^9^ conidia ml^−1^. Three milliliters of the inoculum were added into each pot and mixed well with the soil.

Three experiment variants were designed: first, using soil inoculated with *Trichoderma tomentosum*; second, with *T. ghanense*; and third, with the complex of *T. tomentosum* + *T. ghanense*. The non-inoculated soil of both qualities served as control. Every variant was performed in triplicate. Plant pots were incubated at 23 °C for 14 days in an environmental chamber (Climaslab 470, Barcelona, Spain). The required soil moisture was maintained during all experiments. After 14 days, the chemical properties and CM-cellulase activity of soil were estimated in every pot. Then, all inoculated pots and non-inoculated controls were used to test the effect of indigenous *Trichoderma* species on early seedling growth following standard ISO 11269-2:2012 [43]. Rye seeds (*Secale cereale* L.) with 96% seed germination rate were used throughout this experiment. Twelve seeds of rye per pot were sowed at a depth of 5 mm and were placed in an environmental chamber (Climaslab 470, Barcelona, Spain) at 23 °C applying 16 h illumination period followed by 8 h darkness. After germination, the seedlings were thinned by leaving evenly 7 in every pot. After 14 days, the rye was harvested. The length of roots and shoots was measured, and the dry weight of seedling roots and shoots was estimated.

### 2.7. Statistical Analysis

The data on the root length, shoot length, and dry weight were analyzed using main effects ANOVA with treatment (control, I, II, and III). The soil (control, grassland, and arable soil) was used as categorical predictor. Hotelling’s T^2^ was used for multivariate testing. Thereafter, significant factors were used for ANOVA analysis and Tukey’s HSD for post hoc comparisons. Statistical analysis was performed using PAST3 software. Principal component analysis (PCA) was applied for soil properties. The results were evaluated by numerical methods after data transformation by centering: (x-mean)/stdev. The confidence level was set as *p* < 0.05. Using Microsoft Excel, statistical t values were calculated for determination of variable significance.

## 3. Results

### 3.1. Identification of Trichoderma and Their Physiological Characteristics

Morphological features were used to classify the species of *Trichoderma.* The isolates had fast growing hyaline, later becoming green due to conidium production, colonies, and repeatedly branched conidiophores bearing flask-shaped phialides. According to these morphologic characteristics, the isolates were assigned to *Trichoderma* genus (Figure 1).

Based on phylogenetic analysis of the ITS region and *Tef-1α* gene, these fungal isolates were identified as *T. ghanense* Yoshim. Doi, Y. Abe and Sugiy and *T. tomentosum* Bissett. Comparison of the ITS and TEF1 gene sequences of our *Trichoderma* isolates with sequences of other, closely related *Trichoderma* accessions deposited in GenBank showed significant similarities (99–99.8%) (Table 2 and Table 3).

The phylogenetic analysis was conducted using the maximum parsimony likelihood method and Tamura-Nei model [32]. One of the most parsimonious trees (MP) of each analyzed locus is shown in Figure 2. The consistency index, the retention index for all sites, and length of tree are shown on the phylograms. The percentage (>50%) of replicate trees in which the associated taxa clustered together in the bootstrap test (1000 replicates) is shown next to the branches.

The *Trichoderma* strains had different physiological characteristics. Both *Trichoderma* strains grew at a temperature in the range of 5 to 35 °C (Table 4). At the temperature of 35 °C, *T. ghanense* had higher growth rates than *T. tomentosum*; a significant difference was observed (*p* = 0.002), but at the temperature of 15 °C, the colony of *T. tomentosum* was larger (*p* = 0.00004). Significant differences were observed in colony growth rates of *Trichoderma* spp. at different pH (4, 5, 6, 7, and 8) values. *T. ghanense* was less sensitive to medium pH changers, and *T. tomentosum* grew better in more acidic media.

Moreover, the studied strains demonstrated different abilities for decomposition of organic matter. The results showed that *T. ghanense* was more effective in the decomposition of cellulose (*p* = 0.0002) (Table 4). Its endoglucanase activity was 11.1-fold higher compared to that of *T. tomentosum*; a significant difference was observed (*p* = 0.0001). A statistically significant difference showed that *T. tomentosum* is more suitable for lignin degradation than *T. ghanense* (*p* = 0.0001). There were no significant differences in tyrosinase activity among *Trichoderma* species.

### 3.2. The Chemical Properties and CM-Cellulase Activity of Soil

Soils from different land-use systems (grassland 1S and arable soil 2S) were selected for the experiment (Table 1). Soil chemical analysis showed that grassland 1S was more productive than arable soil 2S, as it was richer (by nearly two-fold) in mobile humic substances with higher content of mobile humic and fulvic acids. Both soils had near-neutral pH (6.92 in the grassland 1S and 6.83 in arable soil 2S), which was favorable for the growth of cereals [44]. The studied grassland 1S was richer in organic carbon, but arable soil 2S showed higher amounts of mobile P_2_O_5_ and mobile K_2_O, which usually depend on the soil use and plants grown.

Before the experiment, the CM-cellulase activity of original soil and the number of fungi were estimated. The CM-cellulase activity of soil was 456.7 ± 3.76 µg GE g^−1^ · dm · 24 h^−1^ in grassland 1S and 100.4 ± 2.56 µg GE g^−1^ · dm · 24 h^−1^ in arable soil 2S, which was 4.5-fold lower than the level in grassland 1S. The number of cultivable fungi was 14.81 × 10_4_ CFU/g in grassland 1S and 5.98 × 10^4^ CFU/g in arable soil 2S. Moreover, no *Trichoderma* spp. were isolated from the soils by the serial dilution method (at dilution 10^3^) before inoculation.

After 14 days of inoculation, the changes in chemical properties of the rhizospheric soil were activated by the fungal inoculum, as shown in Table 5 and Table 6.

Different parameter values of the inoculated soils and controls were uncertain (the differences were not more than 12%), which could be due to the short time of the experiment.

The statistical analysis showed that the properties of the soil inoculated with the complex of *Trichoderma* resembled the original soil features after 14 days of inoculation (Figure 3). The grassland 1S samples inoculated with monocultures were closer to the control soil variant. The analyzed characteristics of all variants of inoculated arable soil 2S were similar to each other and different from the initial and control soil. The selected characteristics of the soils determined 79% of data variability for PCA. The chemical properties of the grassland 1S and arable soil 2S were significantly different. All characteristics that define soil organic matter (soil organic carbon, mobile humic substances, mobile humic and fulvic acids) were strongly intercorrelated with the CM-cellulase activity of the soil.

Inoculation with *Trichoderma* spp. had a significant stimulating effect on soil enzyme activity. The most significant effect on this activity was found in the arable soil 2S when *T. tomentosum* and the complex *T. tomentosum* + *T. ghanense* were applied (Table 5 and Table 6, Figure 3). It was nearly 3-fold higher (2.8-fold and 2.9-fold, respectively) compared to control; at this time, a significant difference was observed.

Surprisingly, we did not observe an expected (as in arable soil 2S) change in the soil properties of grassland 1S. After 14 days of inoculation, the highest CM-cellulase activity of soil was with *T. tomentosum,* but it was only 1.03-fold higher than in the grassland 1S control.

### 3.3. The Evaluation of Rye Seedling Growth

In our experiment, the length of shoot and root of early rye seedlings and their total weight determined the impact of soil on the plant growth promotion ability of different *Trichoderma* strains. The data are presented in Figure 4 and Figure 5. Our study results indicated that the plant growth promotion ability of *Trichoderma* spp. in grassland 1S was different from that in arable soil 2S.

The greatest beneficial effect of *T. tomentosum* on rye was noticed in grassland 1S (Figure 4). A statistically significant difference was established for the length of the roots of rye in grassland with the inoculation of *T. tomentosum* (*p* = 0.005). The total weight of the rye seedling was 3.2% larger in grassland than in non-inoculated soil (control), though a significant difference was not observed (*p* = 0.321). Furthermore, while the weight of the shoot and root of the rye seedlings was larger compared to the control, unfortunately these differences were not significant (*p* = 0.345 and *p* = 0.331, respectively).

As shown in Figure 4, the plant growth promotion ability of *T. ghanense* was detected in arable soil 2S. The total weight of rye seedlings was 19.3% larger in this variant compared to that of the control (significant differences *p* = 0.014). Differences in the length of shoots and roots between inoculated samples and the control were uncertain (no significant differences were observed *p* = 0.241 and *p* = 0.143, respectively) when *T. ghanense* was applied.

The use of inoculum of complex *T. tomentosum* + *T. ghanense* in grassland 1S and arable soil 2S did not have a growth promotion effect on the rye seedlings (Figure 4). The total weight of the rye seedlings was less in grassland 1S compared to the control, and a significant difference was observed (*p* = 0.040). A statistically significant difference was established in arable soil 2S: the shoots were larger in weight (*p* = 0.004)

## 4. Discussion

The successful and efficient use of bioproducts for plant growth promotion depends on the active development of fungi in the substrates. Abiotic factors, as well as biotic, could promote or suppress the action of fungi in the soil. Therefore, the physiological characteristics of fungi and their ability to survive and adapt to various environmental conditions are of great interest. Muniappan and Muthukumar [45] observed the importance of pH values and detected a negative correlation between *Trichoderma koningii* abundance and soil pH. Moreover, Caporale et al. [25] demonstrated the importance of soil quality to the effect of different *Trichoderma* strains on *Brassica rapa* growth.

Our studied fungi had different optimum growth conditions and enzymatic activity. It was observed that *T. tomentosum* could develop better at lower temperatures than *T. ghanense,* but the latter showed optimal growth under a broader range of pH. Moreover, *T. ghanense* was a better decomposer of cellulose, and *T. tomentosum* a better decomposer of lignin. Therefore, we were expecting that their complex would be more viable in different soil qualities and more effective in the mineralization of organic matter than a single strain.

In our experiment, soils with different chemical properties were chosen. The chemical analysis of the studied soils demonstrated the impoverishment of arable soil 2S, as it had less humic substances and organic carbon than grassland 1S. Furthermore, the analysis of the soil enzyme that participates in decomposition of organic matter showed that the CM-cellulase activity was weaker in arable soil 2S compared with grassland 1S. The CM-cellulase activity of soil, as the main cellulolytic enzyme, indicates intensive mineralization of organic matter, which is carried out by microorganisms. The higher the level of CM-cellulase activity, the higher the degree of organic matter mineralization, which provides plants with more nutrients for growth [46]. We hypothesized that the enrichment of soil by fungi with cellulolytic activity could increase CM-cellulase activity and improve the soil with nutrients. Our results demonstrated the dependence of CM-cellulose activity of soil on inoculation with *Trichoderma* spp. The most significant increase in this activity was detected in arable soil 2S, compared with the control, when *T. tomentosum* and the complex *T. tomentosum* + *T. ghanense* were applied. The possible explanation for this might be a lower number of microorganisms and weaker competition with native microorganisms. Grassland microbiome is richer in number and species of microorganisms. Therefore, it was more difficult for *Trichoderma* inoculum to establish its ecological niche. The CM-cellulase activity of soil depends not only on *Trichoderma* activity because other microorganisms also play an important role in this process of cellulose decomposition [44]. As such, interactions between microorganisms can be significant.

Debnath et al. [47] distinguished between the plant growth promoting ability of *Trichoderma* and its role as a biofertilizer, although they did not define strict limits between these two characteristics. The studies on the *Trichoderma* action mechanism showed that it produced active cellulolytic enzymes that resulted in the mineralization of organic matter. At the same time, it could enhance the uptake of nutrients and root hair development.

The promoting effect of fungi such as *Trichoderma* on plant growth is well known and described by different researchers [19,44,48,49]. Application of *Trichoderma* inoculum at an early stage of crop growth permits one to maximize the benefits in terms of root development and nutrient uptake [3]. However, there is not enough data on the behavior of this fungus in different soils and in the natural environment [39]. In nature, a consortium of microorganisms exists in soil, and their interactions could affect the production of different metabolites [21]. These relationships are very important when *Trichoderma* is expected to be used as a plant growth promoter in soil.

In our experiment, the impact of soil on the growth-promoting ability of different *Trichoderma* strains and their complexes was determined by length of shoots and roots of the early rye seedlings and their total weight. Our results showed that the effect of *Trichoderma* inoculation was different in grassland 1S and arable soil 2S and depended on the species of fungi. The beneficial effects of *T. tomentosum* on early rye seedling growth was noted in the grassland (1S), while that of *T. ghanense* was noted in the arable soil (2S).

Root measurements clearly demonstrated the growth-promoting effects of indigenous *Trichoderma* strains. Statistically significant differences were estimated between the rye roots in grassland inoculated with *T. tomentosum* (*p* = 0.005) and roots in the control. Longer roots improve nutrient uptake by the plant and ensure further successful growth. The interaction between plant and *Trichoderma* spp. successfully regulates root architecture and increases the length of lateral and primary roots, resulting in effective nutrient uptake by the plant [20]. *Trichoderma* spp. releases into the rhizosphere auxins, small peptides, volatiles, and other active metabolites that promote root branching and nutrient uptake capacity, thereby boosting plant growth and yield [3]. Several reports have shown that *Trichoderma* spp. colonize plant roots, penetrate the epidermis, and produce various compounds that affect the plant [7,18,44,50]. This could explain the effect of inoculum foremost on the roots of rye in the short-term experiment. Our results showing the different action of indigenous *Trichoderma* strains on rye seedling growth correlated with the established effects of *T. harzianum* and *T. virens* on *Pinus sylvestris* seedlings [19]. Comparing the obtained results with other scientific reports, the differences in the degree of the growth-promoting effect on fungi were noticed. When the experiments were performed in sterile soil, the promoting effect on plant growth was more pronounced. For example, *T. longibrachiatum* increased tomato root volume by 96% [17]. In our experiment, the highest increase in the total weight of early rye seedlings was 19.3% (*p* = 0.014). However, our other findings were consistent with observations by Hajieghrari and Mohammadi [51] that some *Trichoderma* strains capable of colonizing roots showed no significant effect on wheat seedling parameters. Moreover, our results showed that the shoots and roots of the rye seedlings did not always correlate with seedling dry weight. Tančić-Živanov et al. [52] observed similar results with the effect of some *Trichoderma* isolates on early pepper seedlings.

Contrary to our expectations, a growth-promoting effect on early rye seedlings by the complex of *Trichoderma* strains was detected only in arable soil 2S. Though the weight of shoots was larger compared to the control, the total weight of the rye seedlings was not larger.

According to these data, our results contradicted the opinion expressed by Vassilev et al. [21] that microbial co-cultivation could be successfully used as a biofertilizer. These results demonstrated that selection of the strains for use is very important.

However other factors could have an impact on plant growth when the experiment is performed in the soil environment. For example, the humic substances in grassland could be one of the factors that stimulated the growth of seedlings [53,54]. On the other hand, biological and physiological properties such as the competitive abilities of the inoculum are very important, as highlighted in an article published by Singh et al. [44]. Since the dependence of the growth-promoting ability of *Trichoderma* spp. inoculum on the soil type could explain the inefficiency of bio-product in some cases, the selection of a suitable strain of fungi is still of great importance. The complexity of the phenomenon of *Trichoderma* growth-promoting ability requires thorough studies.

## 5. Conclusions

In the present study, the indigenous *Trichoderma* species with different physiological characteristics were tested as biostimulants on the early rye seedlings grown in the sustained grassland and arable soil. The study showed not only that the physiological characteristics of fungi play an important role but also the quality of the soil had an impact on plant growth promotion. Rye seedling root growth was enhanced when *T. tomentosum* was applied in the sustainable grassland (*p* = 0.005), whereas in the arable soil an increase in total weight and shoot weight of the rye seedlings was observed when *T. ghanense* was used (*p* = 0.014 and *p* = 0.024). The expected beneficial effect of *Trichoderma* spp. complex on rye growth promotion was not observed in any tested soil. The obtained results could find application in the development of new and efficient biostimulants and potentially practical strategies for sustainable management.

## Figures and Tables

**Figure 1 jof-08-00085-f001:**
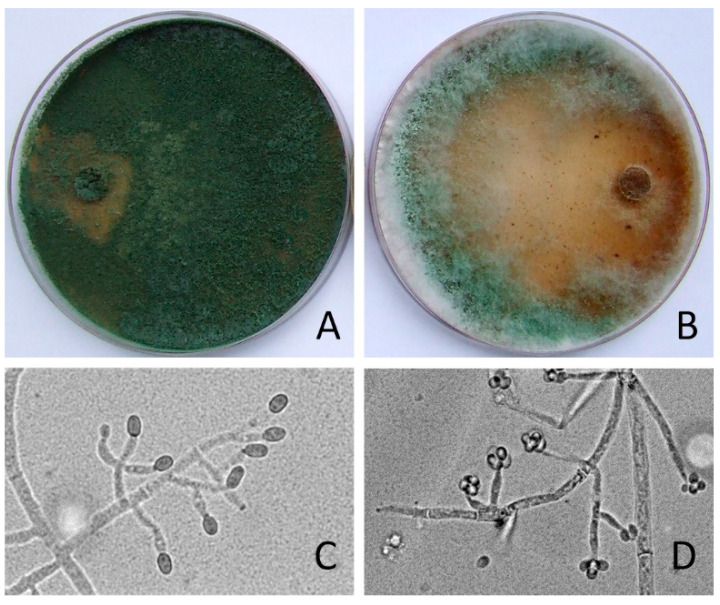
Macromorphology and micromorphology of *Trichoderma* strains on MEA after 7 days. (**A**)—*T. ghanense* colony and (**C**)—conidiophores and conidia. (**B**)—*T. tomentosum* colony and (**D**)—conidiophores and conidia. Magnification × 400.

**Figure 2 jof-08-00085-f002:**
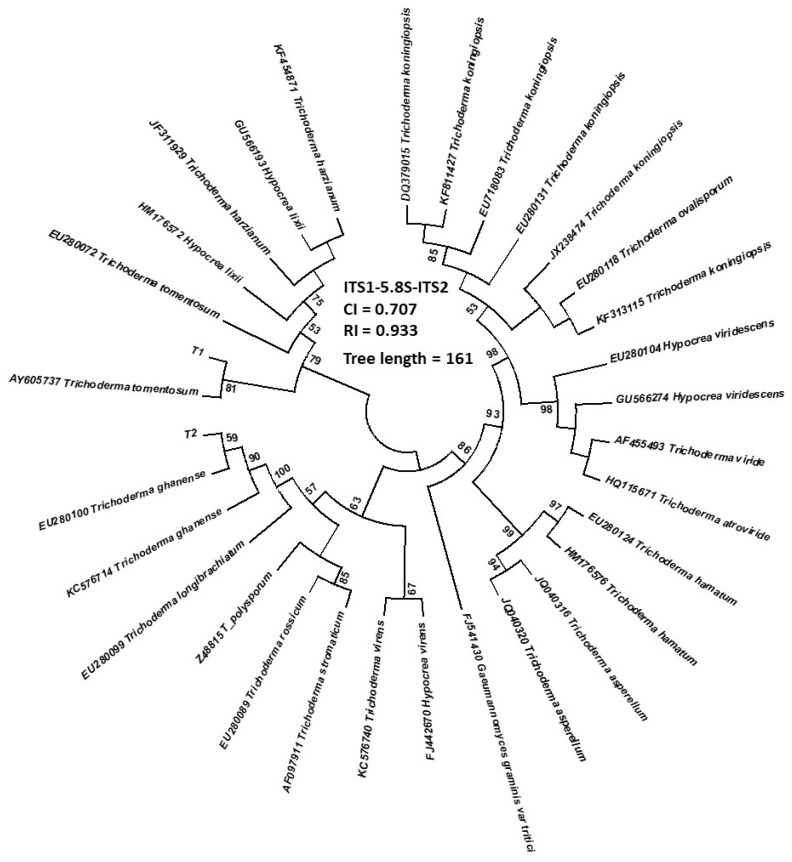
Phylogeny of *Trichoderma* isolates based on ITS1-5,8S-ITS2 and TEF1 loci. The most parsimonious tree for each locus with appropriate length is shown. CI—consistency index, RI—retention index. The percentage >50% of replicate tree is shown next to branches.

**Figure 3 jof-08-00085-f003:**
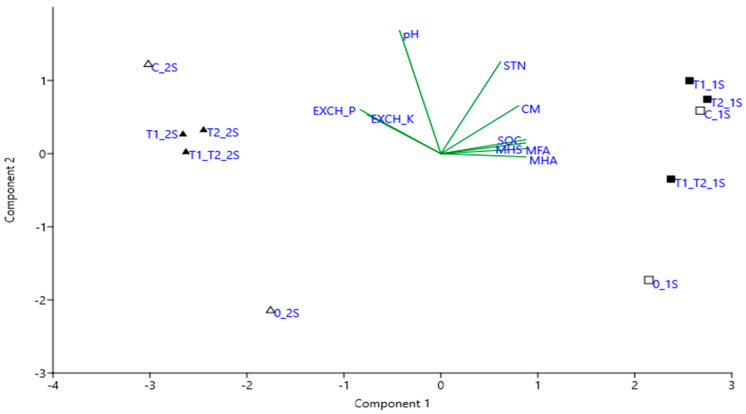
PCA diagrams for variables in experiment on *Trichoderma* spp. inoculated soil originated from different land-use systems. 0_1S—original grassland, C_1S—grassland non-inoculated, T1_1S—grassland inoculated with *T. tomentosum*, T2_1S—grassland inoculated with *T. ghanense*, T1_T2_1S—inoculated with the complex *T. tomentosum* + *T. ghanense*, 0_2S—original arable soil, C_2S—arable soil non-inoculated, T1_2S—arable soil inoculated with *T. tomentosum*, T2_2S—arable soil inoculated with *T. ghanense*, T1_T2_2S—inoculated with the complex *T. tomentosum* + *T. ghanense*. SOC—soil organic carbon, MHA—mobile humic acid, MFA—mobile fulvic acid, MHS—mobile humic substances, CM—CM-cellulase activity of soil, STN—total N, pH—soil pH, EXCH_K—exchangeable K_2_O, EXCH_P—exchangeable P_2_O_5_.

**Figure 4 jof-08-00085-f004:**
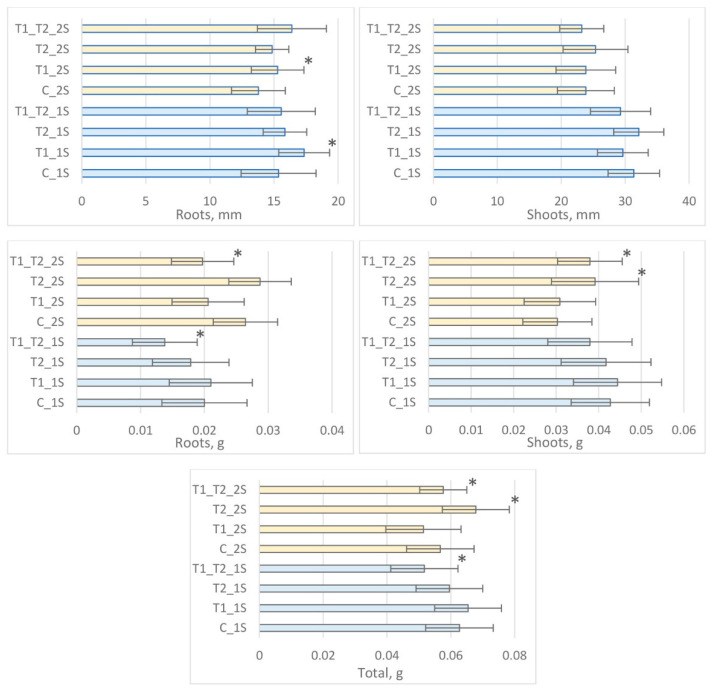
Effects of inoculation with *Trichoderma* spp. on vegetative plant growth. C_1S—grassland non-inoculated, T1_1S—grassland inoculated with *T. tomentosum*, T2_1S—grassland inoculated with *T. ghanense*, T1_T2_1S—grassland inoculated with the complex *T. tomentosum* + *T. ghanense*, C_2S—arable soil non-inoculated, T1_2S—arable soil inoculated with *T. tomentosum,* T2_2S—arable soil inoculated with *T. ghanense*, T1_T2_2S—arable soil inoculated with the complex *T. tomentosum* + *T. ghanense*. * above the bars indicate significant differences.

**Figure 5 jof-08-00085-f005:**
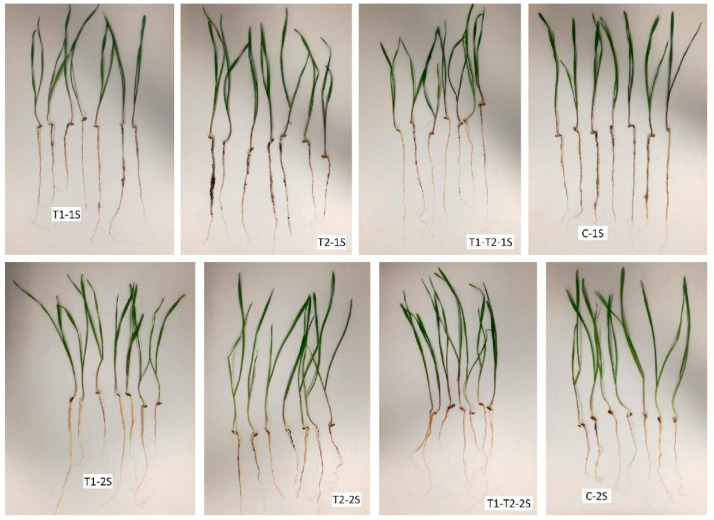
The early seedlings of rye after harvest. T1-1S—grassland inoculated with *T. tomentosum*, T2-1S—grassland inoculated with *T. ghanense*, T1-T2-1S—grassland inoculated with the complex *T. tomentosum* + *T. ghanense*, C-1S—grassland non-inoculated, T1-2S—arable soil inoculated with *T. tomentosum*, T2-2S—arable soil inoculated with *T. ghanense*, T1-T2-2S—arable soil inoculated with the complex *T. tomentosum* + *T. ghanense*, C-2S—arable soil non-inoculated.

**Table 1 jof-08-00085-t001:** Chemical properties of original soil samples.

Chemical Properties	Grassland	Arable Soil	Methods
Soil organic carbon (SOC), %	2.49	1.43	Spectrophotometric measurement method at 590 nm using glucose as a standard after wet combustion [39].
Humus, %	4.30	2.46	Calculated using conversion factor (1.724) from SOC.
Total N (Nt), %	0.115	0.080	Kjeldahl method using a spectrophotometric measurement at 655 nm.
C/N	21.70	17.90	Calculation as ratio of SOC to Nt.
Exchangeable P_2_O_5_, mg/kg	46.90	97.70	A-L method.
Exchangeable K_2_O, mg/kg	98.81	162.90	A-L method.
Labile water-soluble carbon, C_H2O_, g/kg	0.299	0.106	Determined by IR detection after UV-catalyzed persulfate oxidation [40].
Mobile humic substances, (MHS) %	0.383	0.194	Extracted with 0.1 M NaOH [41].
Mobile humic acids, %	0.168	0.092	Determined in 0.1 M NaOH extract.
Mobile fulvic acids, %	0.215	0.102	Determined in 0.1 M NaOH extract.
pH	6.92	6.83	Determined by the potentiometric method in 1 M KCl (1:2.5, *w*/*v*) extract.
CM-cellulase activity, µg GE g^−1^ · dm · 24 h^−1^	456.70	100.40	Determined by the spectrophotometric method [42].

**Table 2 jof-08-00085-t002:** Characterization of *Trichoderma* strains according to ITS region differences.

Strain	Sequence Area Characterization
*T. tomentosum* T1	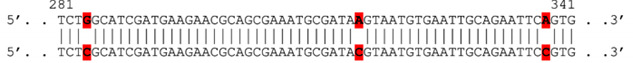
*T. tomentosum* AY605737
*T. ghanense* T2	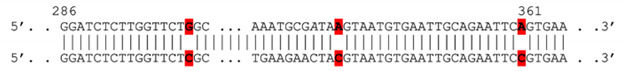
*T. ghanense* EU280100

**Table 3 jof-08-00085-t003:** Characterization of *Trichoderma* strains according to TEF1 gene region differences.

Strain	Sequence Area Characterization
*T. tomentosum* T1	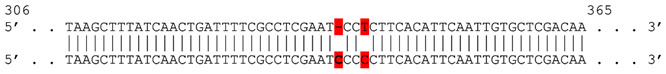
*T. tomentosum* KJ871248
*T. tomentosum* T2	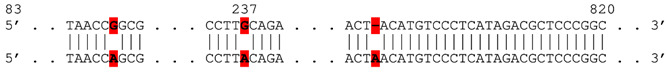
*T. ghanense* EU280043

**Table 4 jof-08-00085-t004:** Physiological characteristics of *Trichoderma ghanense* and *T. tomentosum*. * above the numbers indicate significant differences (*p* < 0.05).

Physiological Characteristics	*T. ghanense*	*T. tomentosum*
Growth (diameter, mm) on PDA at 7 °C	5.0 ± 0.0	5.0 ± 0.0
15 °C	37.0 ± 1.7 *	43.2 ± 2.4 *
26 °C	46.6 ± 2.3 *	45.1 ± 1.7 *
35 °C	56.2 ± 2.3 *	50.0 ± 0.0 *
Growth (diameter, mm) on CzDA at 26 °C and pH 4	55.0 ± 0.0 *	73.7 ± 0.5 *
5	61.0 ± 1.7 *	67.6 ± 0.6 *
6	53.3 ± 2.8 *	63.0 ± 0.0 *
7	51.6 ± 2.8 *	42.0 ± 1.0 *
8	51.0 ± 1.7 *	36.0 ± 0.0 *
Decomposition of cellulose, %	32.7 ± 1.3 *	25.6 ± 0.9 *
Decomposition of lignin, %	11.0 ± 0.8 *	14.2 ± 0.7 *
Enzymatic activity:		
endoglucanase, %	120.6 ± 2.8 *	10.8 ± 1.4 *
peroxidase, a.u./g	106.0 ± 1.4 *	90.0 ± 4.2 *
tyrosinase, c.u./g	0.033 ± 0.0	0.72 ± 0.0
laccase, a.u./g	0.004 ± 0.0	0.004 ± 0.0

**Table 5 jof-08-00085-t005:** The chemical properties of grassland 1S after 14 days of inoculation. Numbers followed by “*” indicate significant differences (*p* < 0.05).

Chemical Properties of Soil	Variants			Control—Non Inoculated Soil
	*T. ghanense*	*T. tomentosum*	*T. ghanense* + *T. tomentosum*	
	1S	1S	1S	1S
SOC, %	2.72 ± 0.02	2.68 ± 0.06	2.74 ± 0.08	2.75 ± 0.18
Humus, %	4.69 ± 0.07	4.61 ± 0.11	4.73 ± 0.11	4.74 ± 0.01
Total N, %	0.208 ± 0.001	0.215 ± 0.003 *	0.207 ± 0.008	0.204 ± 0.004
C/N	13.1	12.5	13.2	13.5
Exchangeable P_2_O_5_, mg/kg	71.90 ± 0.57	71.60 ± 0.00 *	71.60 ± 0.70	69.30 ± 0.28
Exchangeable K_2_O, mg/kg	124.50 ± 0.41 *	128.50 ± 0.13 *	124.20 ± 0.08 *	135.49 ± 0.14
Labile water-soluble carbon, C_H2O_, g/kg	0.268 ± 0.019	0.275 ± 0.026	0.314 ± 0.012	0.291 ± 0.019
Mobile humic substances, %	0.406 ± 0.08	0.434 ± 0.004	0.422 ± 0.014	0.426 ± 0.013
Mobile humic acids, %	0.190 ± 0.001	0.179 ± 0.004 *	0.198 ± 0.004 *	0.188 ± 0.003
Mobile fulvic acids, %	0.216 ± 0.012 *	0.255 ± 0.008	0.224 ± 0.003	0.238 ± 0.004
pH	7.08 ± 0.01 *	7.02 ± 0.01 *	7.02 ± 0.014	7.00 ± 0.00
CM-cellulase activity, µg GE g^−1^ · dm · 24h^−1^	896.00 ± 3.46	930.20 ± 1.91 *	849.40 ± 7.20*	897.60 ± 2.622

**Table 6 jof-08-00085-t006:** The chemical properties of arable soil 2S after 14 days of inoculation. Numbers followed by “*” indicate significant differences (*p* < 0.05).

Chemical Properties of Soil	Variants			Control—Non Inoculated Soil
	*T. ghanense*	*T. tomentosum*	*T. ghanense* + *T. tomentosum*	
	2S	2S	2S	2S
SOC, %	1.39 ± 0.05	1.43 ± 0.01	1.33 ± 0.01	1.38 ± 0.05
Humus, %	2.40 ± 0.06	2.46 ± 0.01	2.30 ± 0.06	2.38 ± 0.11
Total N, %	0.105 ± 0.001	0.109 ± 0.001 *	0.114 ± 0.001	0.123 ± 0.004
C/N	13.2	13.1	11.7	11.2
Exchangeable P_2_O_5_, mg/kg	134.40 ± 1.20	130.60 ± 1.13	129.60 ± 0.14	127.40 ± 2.69
Exchangeable K_2_O, mg/kg	163.40 ± 0.68 *	155.70 ± 0.70 *	149.90 ± 0.68 *	178.30 ± 1.34
Labile water-soluble carbon, C_H2O_, g/kg	0.194 ± 0.015	0.197 ± 0.013	0.182 ± 0.019*	0.199 ± 0.021
Mobile humic substances, %	0.146 ± 0.006	0.151 ± 0.004	0.144 ± 0.006	0.157 ± 0.003
Mobile humic acids, %	0.050 ± 0.001	0.061 ± 0.002	0.051 ± 0.001 *	0.059 ± 0.000
Mobile fulvic acids, %	0.096 ± 0.001	0.090 ± 0.004	0.093 ± 0.004	0.098 ± 0.001
pH	7.13 ± 0.01 *	7.15 ± 0.00	7.13 ± 0.01 *	7.28 ± 0.00
CM-cellulase activity, µg GE g^−1^ · dm · 24 h^−1^	114.80 ± 3.79 *	324.60 ± 3.47 *	332.80 ± 4.30 *	114.50 ± 3.93

## Data Availability

Data available from the corresponding author J.Š. on request.

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
