# Peer review of "Impact of Soil Chemical Properties on the Growth Promotion Ability of *Trichoderma ghanense**, T*. *tomentosum* and Their Complex on Rye in Different Land-Use Systems"

_jof, 2022, doi:10.3390/jof8010085_

Round 1
Reviewer 1 Report
In the present article, the author tried to investigate Impact of Soil on the Growth Promotion Ability of Trichoderma ghanense, T. tomentosum and Their Complex on Rye in Different Land-Use Systems. They got some interesting results. But there were some problems with the manuscript.
- Title. Soil is a complex material, including physical, chemical, biological and other components and properties. So it is unclear what aspect of “soil” in the tile.
In line 15, the author refers to “soil quality”. So what is the difference between “soil” with “soil quality”.
- Add the scientific name of rye in line 15.
- What is “early seedlings” in line 17?
- What are the soil properties in line 22? Physical, chemical, or biological properties?
- For the results in line 26-28, there is a lack of strong data support.
- There is lack of references in line 35-38.
- I think it is not necessary to draw Figure 1 “Design of the experiment” it in the manuscript.
- In Table 2 and Table 3, there were lacks of statistical analysis of data. Compared with the control soil, whether the soil property change induced by Trichoderma inoculation reached significant levels?
- Similarly, whether the rye growth of Trichoderma inoculation treatment were significantly higher than the control treatment in Figure 3?
Thus, the data cannot support the conclusion well in the present manuscript.
Author Response
Dear William Zhang (Assistant Editor) and Reviewers of Journal of fungi,
Re: Revision „Impact of soil on the growth promotion ability of Trichoderma ghanense, T. tomentosum and their complex on rye in different land-use systems“ by Danguolė Bridžiuvienė, Vita Raudonienė, Jurgita Švedienė, Algimantas Paškevičius, Ieva Baužienė, Gintautas Vaitonis, Alvyra Šlepetienė, Jonas Šlepetys, Audrius Kačergius (jof-1499652)
Thank you very much for your kind reviews and comments regarding our manuscript (jof-1499652) entitled above. Now we have carried out revisions according to your comments and hope this will be adequate for the acceptance of this manuscript. Details of corrections according to the comments are as follows;
|
Reviewer #1 |
|
|
Comment |
Title. Soil is a complex material, including physical, chemical, biological and other components and properties. So it is unclear what aspect of “soil” in the title. |
|
Response to Comment |
Thank you very much for your comments. We added the word “environment” as an aspect of physical, chemical, biological factors and conditions that influence the development of organisms. |
|
Comment |
In line 15, the author refers to “soil quality”. So what is the difference between “soil” with “soil quality”. |
|
Response to Comment |
We corrected the sentence as follows; The impact of soil environment on the ability of selected Trichoderma strains and their complex to promote plant growth was determined by the evaluation of the rye (Secale cereale L.) early seedling growth – measuring their length of shoots and roots and as well as their dry weight. |
|
Comment |
Add the scientific name of rye in line 15. |
|
Response to Comment |
Added. |
|
Comment |
What is “early seedlings” in line 17?
|
|
Response to Comment |
We have improved the text in accordance with the recommendation. |
|
Comment |
What are the soil properties in line 22? Physical, chemical, or biological properties? |
|
Response to Comment |
We corrected the sentence as follows; The results indicated a significant (P<0.05) increase in plant growth and the improvement of some soil chemical properties (total N, mobile humic and fulvic acids, soil CM-cellulase activity) in inoculated soils when compared to the control. |
|
General Comment |
For the results in line 26-28, there is a lack of strong data support. |
|
Response to Comment |
We’ve corrected the sentence as follow: The growth of the roots of rye seedlings in the sustained grassland was enhanced when T. tomentosum was applied (p=0.005). There were an increase of total weight and shoots weight of rye seedlings when T. ghanense was used in the arable soil (p=0.014 and p=0.024). The expected beneficial effect of Trichoderma spp. complex on rye growth promotion was not observed in any tested soil. |
|
Comment |
There is lack of references in line 35-38. |
|
Response to Comment |
We now have added the references. |
|
Comment |
I think it is not necessary to draw Figure 1 “Design of the experiment” it in the manuscript. |
|
Response to Comment |
We transfer Figure 1 to supplementary materials. |
|
Comment |
In Table 2 and Table 3, there were lacks of statistical analysis of data. Compared with the control soil, whether the soil property change induced by Trichoderma inoculation reached significant levels. |
|
Response to Comment |
We indicated the significant differences in Table 3, were the soil properties change induced by Trichoderma inoculation. |
|
Comment |
Similarly, whether the rye growth of Trichoderma inoculation treatment was significantly higher than the control treatment in Figure 3? |
|
Response to Comment |
The growth of the roots of rye seedlings in the sustained grassland was enhanced when T. tomentosum was applied (p=0.005). There were an increase of total weight and shoots weight of rye seedlings, in the arable soil when T. ghanense was used (p=0.014 and p=0.024). |
|
Comment |
Thus, the data cannot support the conclusion well in the present manuscript. |
|
Response to Comment |
Conclusion section was rewritten. |
Reviewer 2 Report
The authors investigated the impact of soil on the Growth Promotion Ability of Trichoderma fungi on Rye in Different Land-Use Systems. Very interesting study, but there are some problems on English writing. Please see the details on the comment PDF, and the highlighted sentences need re-write to improve the quality of this article.
In the Introduction section, please provide more details on background of the Trichoderma in agriculture.
For the figures, please provide a high resolution revision.

Author Response
Dear William Zhang (Assistant Editor) and Reviewers of Journal of fungi,
Re: Revision „Impact of soil on the growth promotion ability of Trichoderma ghanense, T. tomentosum and their complex on rye in different land-use systems“ by Danguolė Bridžiuvienė, Vita Raudonienė, Jurgita Švedienė, Algimantas Paškevičius, Ieva Baužienė, Gintautas Vaitonis, Alvyra Šlepetienė, Jonas Šlepetys, Audrius Kačergius (jof-1499652)
Thank you very much for your kind reviews and comments regarding our manuscript (jof-1499652) entitled above. Now we have carried out revisions according to your comments and hope this will be adequate for the acceptance of this manuscript. Details of corrections according to the comments are as follows;
|
Reviewer #2 |
|
|
Comment |
The authors investigated the impact of soil on the Growth Promotion Ability of Trichoderma fungi on Rye in Different Land-Use Systems. Very interesting study, but there are some problems on English writing. Please see the details on the comment PDF, and the highlighted sentences need re-write to improve the quality of this article. |
|
Response to Comment |
Thank you very much for your comments. We have revised the manuscript. |
|
Comment |
In the Introduction section, please provide more details on background of the Trichoderma in agriculture.
|
|
Response to Comment |
Missing information was added to the text. |
|
Comment |
For the figures, please provide a high resolution revision.
|
|
Response to Comment |
We have noticed that the quality of photos in generated pdf file is worse than in the original tiff files. |
Reviewer 3 Report
Dear authors
The research they did is very interesting, it was a pleasure to read work focused on the quality of the soil, however, there are details that can be improved and above all to better understand, then I leave you my general comments
Line 53 "Botrytis" this must be in italics
In the materials and methods section. in isolation, the wording is very simple, it would be convenient to be more explicit in the method of isolation and selection of the fungus (where were they isolated? what type of soil is it? etc.
In the design of the experiment, they did not mention how they did the inoculation, mention the method, for example, was the spore count done? Or was it a plot, maybe that's why they see variability in the results.
In the molecular identification I suggest putting the sequence of the oligos, in addition to including the programs that were used for alignments and phylogenies.
in the results part they do not include the phylogenetic tree, with the markers that they mention. They must include it
They also mention a morphological characterization, however they do not show in the results.
When you show the growth promotion graphs, you do not show the significant differences, and from what is seen in the standard deviations there are no significant differences between treatments. I suggest including the statistic in the chart and switching to a bar chart to make it more understandable.
I suggest reporting a percentage of plant survival
I suggest including photos of the plants, root and shoots
I suggest evaluating whether these trichoderma strains can colonize this plant, as you refer to it in the discussion.
Author Response
Dear William Zhang (Assistant Editor) and Reviewers of Journal of fungi,
Re: Revision „Impact of soil on the growth promotion ability of Trichoderma ghanense, T. tomentosum and their complex on rye in different land-use systems“ by Danguolė Bridžiuvienė, Vita Raudonienė, Jurgita Švedienė, Algimantas Paškevičius, Ieva Baužienė, Gintautas Vaitonis, Alvyra Šlepetienė, Jonas Šlepetys, Audrius Kačergius (jof-1499652)
Thank you very much for your kind reviews and comments regarding our manuscript (jof-1499652) entitled above. Now we have carried out revisions according to your comments and hope this will be adequate for the acceptance of this manuscript. Details of corrections according to the comments are as follows;
|
Reviewer #3 |
|
|
Comment |
Line 53 "Botrytis" this must be in italics
|
|
Response to Comment |
Fixed. |
|
General Comment |
In the materials and methods section. in isolation, the wording is very simple, it would be convenient to be more explicit in the method of isolation and selection of the fungus (where were they isolated? what type of soil is it? etc. |
|
Response to Comment |
We have improved the text in accordance with the recommendation. The “Materials and Methods” 2.1 section is modified to include the details such as time and method. |
|
General Comment |
In the design of the experiment, they did not mention how they did the inoculation, mention the method, for example, was the spore count done? Or was it a plot, maybe that's why they see variability in the results. |
|
Response to Comment |
We have improved the text in accordance with the recommendation. The “Materials and Methods” 2.6 section is modified to include the details such as time and temperature. |
|
Comment |
In the molecular identification I suggest putting the sequence of the oligos, in addition to including the programs that were used for alignments and phylogenies. |
|
Response to Comment |
We have improved the text in accordance with the recommendation. |
|
Comment |
In the results part they do not include the phylogenetic tree, with the markers that they mention. They must include it. |
|
Response to Comment |
We included the phylogenetic tree, with the ITS and TEF1. |
|
Comment |
They also mention a morphological characterization, however they do not show in the results. |
|
Response to Comment |
We have improved the text in accordance with the recommendation. Also, we included the photos of macromorphology and micromorphology of Trichoderma strains. |
|
General Comment |
When you show the growth promotion graphs, you do not show the significant differences, and from what is seen in the standard deviations there are no significant differences between treatments. I suggest including the statistic in the chart and switching to a bar chart to make it more understandable. |
|
Response to Comment |
We do apologize for some of missing data. We included the statistic in the chart. The presentation format of Figure 4 were modified for better understanding. |
|
Comment |
I suggest reporting a percentage of plant survival. |
|
Response to Comment |
We have improved the text in accordance with the recommendation. |
|
Comment |
I suggest including photos of the plants, root and shoots. |
|
Response to Comment |
We included Figure 5 with photos of the plants, root, and shoots. |
|
Comment |
I suggest evaluating whether these trichoderma strains can colonize this plant, as you refer to it in the discussion. |
|
Response to Comment |
We described in the “Discussion” section as follows; The interaction between plant and Trichoderma spp. successfully regulates root architecture, increase the length of lateral and primary root that result in the effectiveness of nutrient uptake by the plant [20]. Trichoderma spp. releases into the rhizosphere auxins, small peptides, volatiles and other active metabolites, which promote root branching and nutrient uptake capacity, thereby boosting plant growth and yield [3]. |
Round 2
Reviewer 1 Report
In the present article, the author made many revises according the reviewer’s comments and improved the manuscript. But there were some problems in the new manuscript.
- In title, soil environment also is a unclear term.
- In Table 4, there were lacks of statistical analysis of data. Among the treatments, whether the indexes of physiological characteristics were changed significantly? Which treatment was the best?
- Similarly, in table 5, there were lacks of standard errors and statistical analysis of data? Why the T. tomentosum can significant improved the total N content?
Author Response
Dear William Zhang (Assistant Editor) and Reviewers of Journal of fungi,
Re: Revision „Impact of soil on the growth promotion ability of Trichoderma ghanense, T. tomentosum and their complex on rye in different land-use systems“ by Danguolė Bridžiuvienė, Vita Raudonienė, Jurgita Švedienė, Algimantas Paškevičius, Ieva Baužienė, Gintautas Vaitonis, Alvyra Šlepetienė, Jonas Šlepetys, Audrius Kačergius (jof-1499652)
Thank you very much for your kind reviews and comments regarding our manuscript (jof-1499652) entitled above. Now we have carried out revisions according to your comments and hope this will be adequate for the acceptance of this manuscript. Details of corrections according to the comments are as follows;
|
Comment |
In title, soil environment also is a unclear term. |
|
Response to Comment |
In title we have replaced “soil environment” with “soil chemical properties”. |
|
Comment |
In Table 4, there were lacks of statistical analysis of data. Among the treatments, whether the indexes of physiological characteristics were changed significantly? Which treatment was the best? |
|
Response to Comment |
We indicated the significant differences in Table 4. T. ghanense was the best because of its higher cellulose decomposition ability and steady growth over wide range of soil pH. |
|
Comment |
Similarly, in table 5, there were lacks of standard errors and statistical analysis of data? Why the T. tomentosum can significant improved the total N content? |
|
Response to Comment |
We checked all the data in Table 5, the standard error values were added for Table 5. We think T. tomentosum did not inhibit nitrogen fixing bacteria and promoted their function. El-Katatny hypothesized that Trichoderma and Azospirillum could have synergistic effect. However, more detailed research is needed.
El-Katatny M.H. Enzyme production and nitrogen fixation by free, immobilized and coimmobilized inoculants of Trichoderma harzianum and Azospirillum brasilense and their possible role in growth promotion of tomato. Food Technol. Biotechnol., 2010, 48(2): 161-174. Elmerich, C., & Newton, W. E. (Eds.). (2007). Associative and Endophytic Nitrogen-fixing Bacteria and Cyanobacterial Associations. Nitrogen Fixation: Origins, Applications, and Research Progress. doi:10.1007/1-4020-3546-2. Contreras-Cornej, H.A, Macías-Rodríguez L., Ek del-Val, Larsen J., Ecological functions of Trichoderma spp. and their secondary metabolites in the rhizosphere: interactions with plants. FEMS Microbiology Ecology, 2016, 92(4), fiw036. |